# Plastid phylogenomics of Pleurothallidinae (Orchidaceae): Conservative plastomes, new variable markers, and comparative analyses of plastid, nuclear, and mitochondrial data

**Anna Victoria Silvério R. Mauad**[1]*, **Leila do Nascimento Vieira**[1], **Valter Antônio de Baura**[2], **Eduardo Balsanelli**[2], **Emanuel Maltempi de Souza**[2], **Mark W. Chase**[3,4], **Eric de Camargo Smidt**[1]*

**1** Departamento de Botânica, Universidade Federal do Paraná, Curitiba, Paraná, Brazil, **2** Departamento de Bioquímica e Biologia Molecular, Universidade Federal do Paraná, Curitiba, Paraná, Brazil, **3** Royal Botanic Gardens, Kew, Richmond, Surrey, United Kingdom, **4** Department of Environment and Agriculture, Curtin University, Perth, Western Australia, Australia

* annavmauad@gmail.com (AVSRM); ecsmidt@gmail.com (ECS)

**Data Availability Statement:** All DNA sequences generated are available from the GenBank

## Abstract

We present the first comparative plastome study of Pleurothallidinae with analyses of structural and molecular characteristics and identification of the ten most-variable regions to be incorporated in future phylogenetic studies. We sequenced complete plastomes of eight species in the subtribe and compared phylogenetic results of these to parallel analyses of their nuclear ribosomal DNA operon (26S, 18S, and 5.8S plus associated spacers) and partial mitochondrial genome sequences (29–38 genes and partial introns). These plastomes have the typical quadripartite structure for which gene content is similar to those of other orchids, with variation only in the composition of the *ndh* genes. The independent loss of *ndh* genes had an impact on which genes border the inverted repeats and thus the size of the small single-copy region, leading to variation in overall plastome length. Analyses of 68 coding sequences indicated the same pattern of codon usage as in other orchids, and 13 protein-coding genes under positive selection were detected. Also, we identified 62 polymorphic microsatellite loci and ten highly variable regions, for which we designed primers. Phylogenomic analyses showed that the top ten mutational hotspots represent well the phylogenetic relationships found with whole plastome sequences. However, strongly supported incongruence was observed among plastid, nuclear ribosomal DNA operon, and mitochondrial DNA trees, indicating possible occurrence of incomplete lineage sorting and/or introgressive hybridization. Despite the incongruence, the mtDNA tree retrieved some clades found in other analyses. These results, together with performance in recent studies, support a future role for mitochondrial markers in Pleurothallidinae phylogenetics.

database. GenBank accession numbers of all species sampled are provided in S1 Table. Alignment matrices and trees are available from the TreeBase database (accession 27774). All other relevant data are within Supporting Information files.

**Funding:** AVSRM - 88887.311411/2018-00 - Coordenação de Aperfeiçoamento de Pessoal de Nível Superior - https://www.gov.br/capes/pt-br ECS - 311001/2014-9 and 203304/2018-7 - Conselho Nacional de Desenvolvimento Científico e Tecnológico - https://www.gov.br/cnpq/pt-br The funders had no role in study design, data collection and analysis, decision to publish, or preparation of the manuscript.

**Competing interests:** The authors have declared that no competing interests exist.

## 1. Introduction

Neotropical Pleurothallidinae (Epidendreae, Epidendroideae) are the largest orchid subtribe, comprising more than 5,000 accepted species in 44 genera [1, 2]. These are mostly epiphytes and can occupy almost all habitat types from North America (Florida) and the Caribbean through southern South America (Argentina) [3], although most pleurothallid species have narrow endemic distributions, and, therefore, many are considered endangered (e.g. [4]).

Previously, circumscriptions of Pleurothallidinae genera were based on morphological characters, and so was the inference of evolutionary relationships: Luer [5] classified them into informal groups or "affinities", relying on anther position and presence/absence of the annulus (a ring-like abscission zone on leaves). However, the first reclassification of Pleurothallidinae based on molecular evidence [6] highlighted several problems at generic and infrageneric levels (e.g. the polyphyletic "supergenus" *Pleurothallis* R.Br.), but some of the taxonomic changes that followed [7] were contested by Luer [8] due to the lack of morphological correlates, sampling problems, and the relatively low numbers of molecular markers used (i.e. often just nuclear ribosomal internal transcribed spacer, nrITS, plastid *matK* gene, and *trnL-trnF* intron/intergenic spacer).

This taxonomic controversy inspired more phylogenetic studies in the subtribe, mostly focused on specific genera and based almost only on nrITS (e.g. [9–17]). These studies initiated another round of reclassification in Pleurothallidinae [18], in which phylogenetic positions and generic classification were reassessed, providing a good framework for future studies. Karreman's proposal [18] recognized nine genera affinities, but due to a large number of species and infrageneric categories in the subtribe, some relationships remained inconclusive because the compiled phylogenetic trees did not sample and fully resolve all Pleurothallidinae clades. Thus, some nomenclatural instability in the subtribe has continued and greatly affects regulations on international trade and conservation efforts that depend on Red Lists and population genetic studies.

In addition to nrITS, plastid genomes (plastomes) have been a source of good markers for Orchidaceae at various taxonomic levels [19]. In Pleurothallidinae, the *matK* gene and *trnK*−UUU intron are the main plastid markers used in combination with nrITS, but they are insufficiently variable to generate well resolved relationships for many genera. More recent molecular studies in the subtribe have used a wider number of plastid DNA markers (i.e. *ycf1* [20], *matK*, *psbD-trnT*, *rps16-trnQ*, *trnH-psbA*, and *trnS-trnG* intergenic spacers [21]) in combination with nrITS, which improved tree resolution and support. However, the general utility of plastid DNA markers for the subtribe is still under-investigated, particularly which are the most variable and informative.

The advent of next-generation sequencing (NGS) has made plastome sequencing faster and more accessible [22], and they have become the main source of phylogenetic information for angiosperms [19, 23]. In Orchidaceae, plastome sequences have been compared among genera and species to find the most-variable regions, termed mutational hotspots [24–29]. Hence, comparative analyses of complete plastome sequences of Pleurothallidinae can indicate better molecular markers, but thus far only three plastomes of the subtribe are publicly available. Only the overall structure of these plastomes was analyzed and compared so far [30], with no analysis of which regions were more variable.

With this in view, we performed the first plastome study for Pleurothallidinae to identify mutational hotspots for use in future phylogenetic and population studies. We sequenced eight Pleurothallidinae plastomes and included two of those previously published in our analyses. We analyzed genome size and structure, gene content and order, and inverted repeats borders. We also compared codon usage and frequency and detected protein-coding genes under

positive selection. In addition, we identified polymorphic microsatellite loci and the ten most-variable regions, for which we designed primers. Finally, we used a maximum likelihood phylogenetic approach to evaluate and compare relationships of pleurothallid genera [18] based on these plastomes, nuclear ribosomal DNA operons, mitochondrial data, and combined analyses.

## 2. Materials and methods

### Species sampling

We included eight Pleurothallidinae species from individuals cultivated in the greenhouse of the Botany Department of Universidade Federal do Paraná (UFPR), Brazil. Fresh leaves were collected from these individuals for DNA extraction and NGS, and the vouchers were deposited at the UFPR herbarium (UPCB) [31]. We also used the plastome sequence of *Anathallis obovata* (Lindl.) Pridgeon & MW.Chase (MH979332), which was previously published by us [30], and obtained from GenBank the plastome sequence of *Masdevallia picturata* Rchb.f. (KJ566305), totaling ten species from subtribe Pleurothallidinae, representing seven of the nine generic affinity groups proposed by Karremans [18]. We also downloaded from GenBank the plastome sequences of two Laeliinae species to serve as the outgroup, according to the most recent classification of Orchidaceae [2]. Voucher and GenBank accession numbers of all species sampled are provided in S1 Table.

### DNA extraction and NGS

We followed a plastid-enrichment procedure using 2 g of leaf tissue according to Sakaguchi *et al*. [32] and then extracted genomic DNA following the Doyle & Doyle [33] protocol, with reagent scaling to 2 mL microtubes and increasing the incubation time to 2−4 hours at 60˚C. DNA was purified with DNA Clean and Concentrator kit (Zymo Research, Orange, CA) and sequenced on an Illumina MiSeq® using DNA Nextera XT Sample Prep kit (Illumina™) and MiSeq Reagent Kit V2 (Illumina™).

### Genome assembly

The paired-end reads obtained from Illumina MiSeq sequencing ($2 \times 250$ bp) were trimmed at 0.05 error probability limit and discarded when below 50 bp long with the CLC Genomics Workbench 8.0 (CLC Bio, Qiagen). Reads were then used in genome assembly with a mixed guided and *de novo* approach, performed in both CLC Genomics Workbench and Geneious Prime 2020.0.5 (Biomatters Ltd.). In the latter, we generated contigs from a *de novo* assembly of the reads using the MIRA 4.0 plugin, with the most accurate settings. In CLC Genomics Workbench, we mapped the reads in the complete genome sequence of *Anathallis obovata* (MH979332) using the default configuration and then generated a consensus sequence that was gapped in low-coverage areas ($\leq 5\times$). These gaps were manually filled using the contigs, editing the former consensus sequence into a new one. The reads were mapped into the new consensus sequence with CLC Genomics Workbench, and the entire map was visually inspected to check for mismatches and assembly errors, which were manually corrected using the contigs. Sequencing information for each sample is available in S2 Table.

**Genome annotation and graphical representations.** Gene, coding sequence (CDS), ribosomal DNA (rDNA), and transport RNA (tRNA) annotations were imported from the complete plastome sequence of *Anathallis obovata* [30] in Geneious Prime. All annotations were manually verified and edited against those of the *Dendrobium officinale* Kimura & Migo (KC771275) and *Masdevallia picturata* (KJ566305) reference genomes. The IRs were identified

and annotated using the Find Repeats tool of Geneious Prime and verified through REPuter online version (http://bibiserv.cebitec.uni-bielefeld.de/reputer). All sequences were submitted to GenBank through Geneious Prime. The.gb files of Pleurothallidinae plastomes were uploaded to Organellar Genome DRAW v1.1 [34] to make the genome maps and to IRscope [35] to produce the graphical representation of IR/LSC and IR/SSC junctions.

**Mitochondrial and nuclear sequences.** From the sequencing output of the eight species sequenced here plus that of *Anathallis obovata* sequenced by us previously [30], we recovered the complete sequence of the nuclear ribosomal DNA (nrDNA) operon (26S, 18.S, and 5.8S plus the associated spacers, ITS1 and ITS2) and parts of the mitochondrial genome (S2 Table). We mapped the raw reads to the nuclear ribosomal DNA sequence of *Phalaenopsis japonica* (Rchb.f.) Kocyan & Schuit. (MN221419) and the complete mitochondrial genome of *Allium cepa* L. (NC_030100.1) with Geneious Prime, using BBmap, with the highest sensitivity and Kmer length = 8. Also, in Geneious Prime, we annotated the nrDNA operon based on *Acianthera luteola* (Lindl.) Pridgeon & M.W. Chase (KX495754) and using BLAST. For mitochondrial DNA (mtDNA), we extracted only regions with coverage depth ≥ 8× as consensus sequences and imported CDSs, rDNAs, and tRNAs annotations from the reference sequence (NC_030100.1), which were then verified manually (S3 Table). Consensus sequences without annotations were excluded from the analyses and GenBank submission.

## Sequence alignments

All alignments were made in Geneious Prime. We performed a Mauve alignment of the ten complete Pleurothallidinae plastomes with the progressiveMauve algorithm [36] to compare their general structure. For all subsequent analyses, we removed one of IRs from the plastome sequences to avoid overrepresentation. All alignments were made using MAFFT v.7.450 [37] with the FFT-NS-2 algorithm.

## Codon usage and molecular evolution analyses

Codon usage and molecular evolution analyses were performed in the R software environment (https://www.r-project.org/). For the codon usage analysis, we extracted all complete CDS annotations from each of the ten Pleurothallidinae plastomes using Geneious Prime tools. Relative synonymous codon usage (RSCU) and codon frequencies were calculated for each CDS set using the SeqinR package [38]. For the molecular evolution analysis, we extracted and aligned separately all CDSs in common to the ten Pleurothallidinae plastomes in Geneious Prime. This analysis consisted of the application of Tajima's D neutrality test [39], from pegas package [40], in each CDS alignment. All codes and datasets used are available in S1 File.

## Simple sequence repeats (SSRs)

Plastid SSRs were identified for the ten Pleurothallidinae plastomes through MISA-web online program [41], using the following search minimum parameters: ten repetitions for mononucleotide motifs, five repetitions for dinucleotide motifs, and three repetitions for tri-, tetra-, penta-, and hexanucleotide motifs. All SSRs were manually annotated to the sequences, which were then aligned (S2 File). We designed primers for polymorphic SSRs present in at least seven plastomes using Geneious Prime, with the following characteristics: 18−27 bp in length, guanine-cytosine (GC) content between 20−80%, melting temperature (Tm) of 57−63°C with a maximum variation of 1°C between primer pairs, and product sizes between 100−500 bp.

## Sequence variability and indels events

We aligned the ten Pleurothallidinae plastome sequences and then extracted all introns and intergenic spacers (IGSs) with 150 bp minimum length using Geneious Prime tools. We also extracted molecular markers that have been commonly used in Orchidaceae phylogenetics, such as the *trnH-psbA* intergenic spacer, the *matK* CDS, and the 3' portion of *ycf1* CDS [19, 42]. All aligned sequences were uploaded to DnaSP v.6 software [43] to obtain the total number of variable sites and insertions/deletions (indels). These data were used to calculate the sequence variability (SV) *sensu* Shaw *et al.* [44] but considering indels as events instead of sites to reduce homoplasy in these alignments [45]. Therefore, we used the following Eq (1) to calculate SV, where $l$ = total length in bp, $m$ = total number of mutations, and $i$ = number of indels events.

$$SV = \frac{m + i}{l + m + i} \times 100 \qquad (1)$$

The ten sequences with the highest SV were selected as potential molecular markers for Pleurothalldinae. We designed primers for these sequences in Geneious Prime with the following characteristics: 18−27 bp in length, GC content between 20−80%, 57−63°C Tm, maximum variation of 2°C annealing temperature between primer pairs, and product size between 100 −1,000 bp.

We also extracted the IR, LSC, and SSC regions of each Pleurothallidinae plastome using Geneious Prime tools and performed multiple and pairwise alignments using *Masdevallia picturata* as the reference. Back in DnaSP, we computed the number of indels events per region and each plastome.

## Phylogenetic analyses

All phylogenetic analyses were performed with maximum likelihood (ML) using IQ-tree v.1.6.11 [46], with 1,000 ultra-fast bootstrap replicates and *-bnni* strategy to reduce the risk of overestimation [47−49]. The best nucleotide substitution model was set for each dataset under the AIC criterion using ModelFinder [50], implemented on IQ-tree. The resulting trees were visualized and edited using Figtree v.1.4.1 (*http://tree.bio.ed.ac.uk*) and CorelDRAW X8 (*https://www.coreldraw.com/*). We analyzed and compared the following aligned datasets including the twelve Epidendreae samples: complete plastome sequences (one IR), plastid CDSs, plastid non-coding sequences, and top ten plastid mutational hotspots. We also analyzed, compared, and combined the following datasets including nine Pleurothallidinae samples (excluding *Masdevallia picturata*): complete plastome sequences (one IR), mtDNA, and nrDNA operon. For partitioned datasets, we first defined the best substitution model for each partition before combining them using Geneious Prime tools. Bootstrap percentages (BP) above 95 were considered strongly supported [51]. The Epidendreae complete plastome dataset and the Pleurothallidinae combined dataset are available in S3 File.

## 3. Results and discussion

The overall characteristics of the plastomes analyzed are summarized in Table 1 and those of the mitochondrial DNA in Table 2. Pleurothallidinae plastomes varied from 148,246 to 157,905 bp in length, with *Acianthera recurva* (Lindl.) Pridgeon & M.W. Chase the shortest and *Myoxanthus exasperatus* (Lindl.) Luer the longest. GC content varied little (36.9−37.1%). They all possess the typical quadripartite structure of most angiosperms plastomes: LSC of 83,694−85,605 bp and SSC of 10,573−18,444 bp, interspersed by two IRs of 25,242−27,020 bp (S1 Fig) [52].

**Table 1. General characteristics of Pleurothallidinae plastomes analyzed.**

| Taxon | Length (bp) | LSC (bp) | SSC (bp) | IR (bp) | %GC |
|---|---|---|---|---|---|
| *Acianthera recurva* | 148,246 | 84,871 | 10,573 | 26,401 | 37.0 |
| *Anathallis microphyta* | 154,558 | 84,597 | 15,993 | 26,984 | 37.0 |
| *Anathallis obovata* | 155,515 | 86,694 | 17,923 | 26,949 | 37.1 |
| *Dryadella lilliputiana* | 156,807 | 84,943 | 17,992 | 26,936 | 37.1 |
| *Masdevallia picturata* | 156,045 | 84,948 | 18,029 | 26,534 | 36.9 |
| *Myoxanthus exasperatus* | 157,905 | 85,605 | 18,260 | 27,020 | 37.1 |
| *Octomeria grandiflora* | 155,284 | 84,916 | 17,874 | 26,247 | 36.9 |
| *Pabstiella mirabilis* | 150,317 | 83,699 | 16,134 | 25,242 | 37.1 |
| *Stelis grandiflora* | 157,535 | 85,205 | 18,444 | 26,943 | 36.9 |
| *Stelis montserratii* | 157,479 | 85,147 | 18,366 | 26,983 | 36.9 |

Gene composition is similar among the Pleurothallidinae plastomes analyzed: 102 genes are shared, being 68 CDSs, 4 rDNAs, and 30 tRNAs (S4 Table), and a variable set of *ndh* genes (S2 Fig). Of the 11 *ndh* genes, only *ndhB*, *ndhD*, and *ndhH* are present in all plastomes, but those with complete reading frames include only *Myoxanthus exasperatus*, *Octomeria grandiflora* Lindl., *Stelis grandiflora* Lindl., and *S. montserratii* (Porsch) Karremans. The *Masdevallia picturata* and *Octomeria grandiflora* plastomes are the only ones that have all 11 *ndh* genes intact, whereas *Acianthera recurva* has the fewest of these genes, only five, none with complete CDSs. Despite the variation observed in *ndh* gene content, the Mauve alignment showed no inversions or other rearrangements (S3 Fig). Complete losses and pseudogenization of *ndh* genes are common in Orchidaceae and have occurred independently [53–58]. These events have been linked to IR/SSC border instability and IR expansion [54, 57, 58], which we observed for the IR borders among these Pleurothallidinae plastomes (Fig 1). The IRa/SSC junction (JSA) is located within the *ycf1* gene, producing a partial copy of this gene in IRb, but there are three types of IRb/SSC junctions (JSB) in Pleurothallidinae that can be classified into types I, II, and III *sensu* Luo *et al.* [25]. Type I occurs in *Anathallis obovata*, *Dryadella lilliputiana* (Cogn.) Luer, and *Pabstiella mirabilis* and is characterized by the presence of *ndhF* in the SSC without overlapping the JSB. Type II was the most common and detected in *Anathallis microphyta*, *Masdevallia picturata*, *Myoxanthus exasperatus*, *Octomeria grandiflora*, *Stelis grandiflora*, and *S. montserratii*, in which *ndhF* overlaps the JSB. Finally, type III, *ndhF* deletion, was found only in *Acianthera recurva*.

We also observed a significant reduction of the *ycf1* partial copy in *Pabstiella mirabilis* due to a 1,000 bp shift of the gene into the SSC. Size reduction of *ycf1* in IRs was highlighted as one

**Table 2. General characteristics of mtDNA dataset for Pleurothallidinae.**

| Taxon | Total length (bp) | %GC | # genes | # CDSs | # tRNAs | # rDNAs |
|---|---|---|---|---|---|---|
| *Acianthera recurva* | 62,237 | 48.3 | 38 | 26 | 9 | 3 |
| *Anathallis microphyta* | 32,417 | 46.5 | 32 | 25 | 4 | 3 |
| *Anathallis obovata* | 46,959 | 47.0 | 33 | 25 | 5 | 3 |
| *Dryadella lilliputiana* | 83,225 | 47.2 | 36 | 25 | 8 | 3 |
| *Myoxanthus exasperatus* | 61,363 | 47.4 | 37 | 26 | 8 | 3 |
| *Octomeria grandiflora* | 63,381 | 48.2 | 33 | 25 | 6 | 2 |
| *Pabstiella mirabilis* | 43,635 | 48.4 | 29 | 25 | 1 | 3 |
| *Stelis grandiflora* | 49,861 | 47.4 | 34 | 25 | 6 | 3 |
| *Stelis montserratii* | 73,468 | 47.4 | 38 | 26 | 9 | 3 |

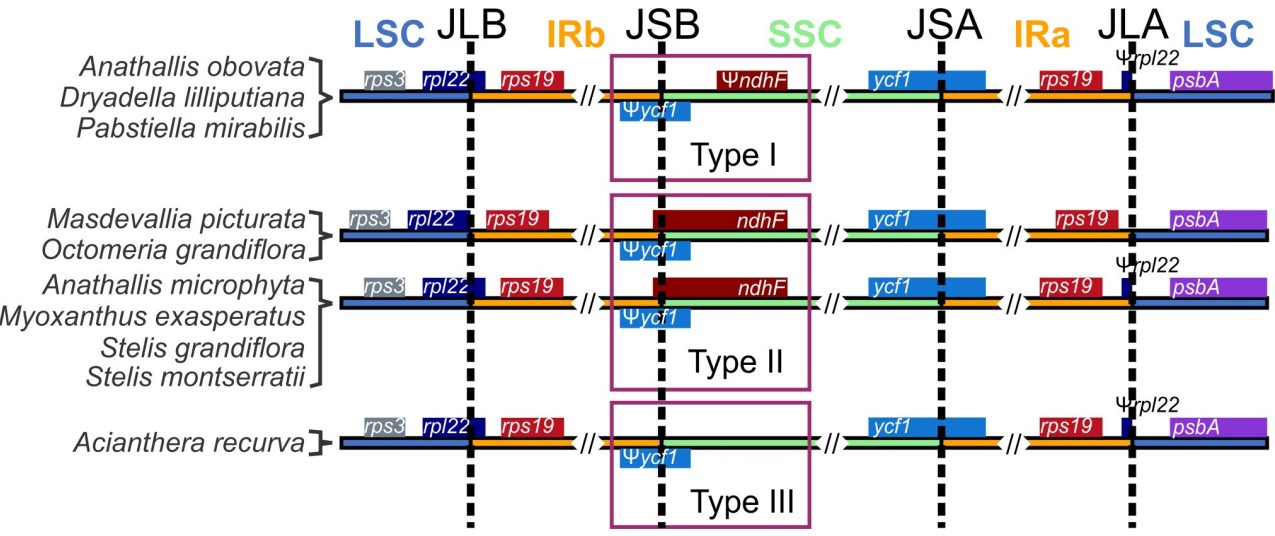

**Fig 1. Inverted repeats (IRs) borders of the ten Pleurothallidinae plastomes analyzed.** IRs/SSC junction types *sensu* Luo *et al*. [25].

possible outcome of IRs/SSC junction instability promoted by deletion/retention of *ndh* genes, especially *ndhF* [54]. *Pabstiella mirabilis* has only 15 bp of *ndhF*, but other Pleurothallidinae with this gene missing or pseudogenized did not suffer alterations in *ycf1* size at the IRs. Thus, IR/SSC junction instability appears to have many attributes not exclusively dependent on the *ndh* gene content. The specific complement of *ndh* genes in *P. mirabilis* could have produced some IR reduction, but this was too small to alter gene content.

The LSC/IRb junction (JLB) varied among Pleurothallidinae as well. In *Octomeria grandiflora* and *Masdevallia picturata*, the JLB is situated in the *rpl22-rps19* IGS and *rps19* gene, respectively, producing a partial copy of the *rps19* gene at IRa in this case (Fig 1). However, in the other Pleurothallidinae, the JLB is located in the *rpl22* gene, producing a partial copy of this gene in IRa (Fig 1). We believe that retention of complete CDSs of all 11 *ndh* genes in *Masdevallia picturata* and *Octomeria grandiflora* may have caused a slight IR expansion, thus generating the variation observed in JLB position in these two plastomes relative to the others. Despite this, all ten plastomes have the *trnH*-GUG and *rps19* genes in both IRs, even though *rps19* is truncated in the IRa of *Masdevallia picturata*, which is the type III JLB, conserved among monocots [59].

IR expansion/retraction is considered one of the main causes of length variation in angiosperm plastomes [60–62]. However, the first study in *Dendrobium* Sw. showed a major contribution of the LSC to plastome total size variation due to the presence of large number of indels in this region [27]. This same large LSC contribution was also observed in Pleurothallidinae: of the 2,417 indel events, 1,899 (78.57%) are in the LSC, 384 (15.89%) in the SSC, and 134 (5.54%) in the IR (S5 Table). Nonetheless, when we analyzed indels events in each plastome, we found no correlation between the number of LSC indels and genome size ($R = −0.52$, $p = 0.16$), but we observed such a correlation with indels in the IR ($R = −0.69$, $p = 0.04$). We also found a correlation between genome size and LSC size ($R = 0.70$, $p = 0.04$), SSC size ($R = 0.93$, $p < 0.01$), and *ndh* CDSs ($R = 0.68$, $p = 0.05$). SSC size, in turn, is sensitive to *ndh* gene content ($R = 0.66$, $p = 0.05$) and their CDSs ($R = 0.77$, $p = 0.01$). Correlation plots are presented in S4 Fig. These results contradict, in part, the findings of Niu *et al*. [27] but also show that plastome size is affected by a combination of factors, for which the relative importance varies among clades. For Pleurothallidinae plastomes, which have few generic differences,

indels in the most conservative region, the IR, have a greater impact on length than LSC indels, despite their predominance in the latter. Also, the sizes of LSC and SSC contribute unequally to Pleurothallidinae genome size because both varied more than the IR, but SSC size varied the most: about four times the range observed for LSC and IR (7,871 bp, against 1,906 bp of LSC and 1,778 bp of IR). In addition, seven of the 11 *ndh* genes are located in SSC, so it is not surprising that *ndh* gene composition greatly influences the length of this region and, consequently, plastome size because it is the main distinctive feature among Pleurothallidinae genomes. This influence of *ndh* gene deletion/retention on the SSC size has been widely reported in the literature [53, 54, 63, 64], but the importance of this region for orchid plastome size variation has only been observed for Pleurothallidinae and *Bulbophyllum* Thouars [29] thus far.

## Protein-coding genes

We observed that relative codon frequencies are similar among Pleurothallidinae (Fig 2, S6 Table). The most common were *AAA* (lysine) in *Acianthera recurva*, *Anathallis microphyta*, and *Pabstiella mirabilis*, *GAA* (glutamate) in *Dryadella lilliputiana*, and *ATT* (isoleucine) in the remaining species, whereas *TGC* (cysteine) was the rarest after stop codons. Among amino acids, leucine was the most frequent, and cysteine the least, similar to that observed in *Bulbophyllum* plastomes [29]. *ATG* (methionine) was the main start codon, but there is *GTG* in *rps19*, *ACG* in *ndhD* and *rps2*, *CTG* in *ndhC* (just *Dryadella lilliputiana*), and *ATT* in *matK*, except in *Pabstiella mirabilis*, which has an alternative start codon [65]. For relative synonymous codon usage, we identified a preference for codons that end in A/T (AT$_3$) instead of G/C (GC$_3$), a bias also observed for other angiosperms, including *Bulbophyllum*, and likely correlated with the high AT content of plastomes [29, 66–68].

To assess selection in protein-coding genes, we applied Tajima's D test in the 68 CDSs shared among Pleurothallidinae. The results indicated that 13 genes are under positive selection: *accD*, *atpB*, *petB*, *psbB*, *psbT*, *rbcL*, *rpl22*, *rpl32*, *rpl33*, *rpoC1*, *rps18*, *ycf1*, and *ycf2* (S7 Table). These genes have point mutations (SNPs), alignment gaps, and size variation. One of the most informative markers for land plants, *ycf1* is the second largest gene in the plastome and has been suggested as a potential DNA barcode [42]. The *accD*, *rbcL*, *rpl22*, *rpl32*, *rpoC1*, and *ycf2* genes also appear to be under positive selection in other Orchidaceae, perhaps related to their adaptative capacities [29, 68, 69].

## Microsatellites (SSRs)

Simple sequence repeats (SSRs), also known as short tandem sequences or microsatellites, are 1–6 bp repeats common in all genomes [70, 71]. We identified 1,290 SSRs in Pleurothallidinae,

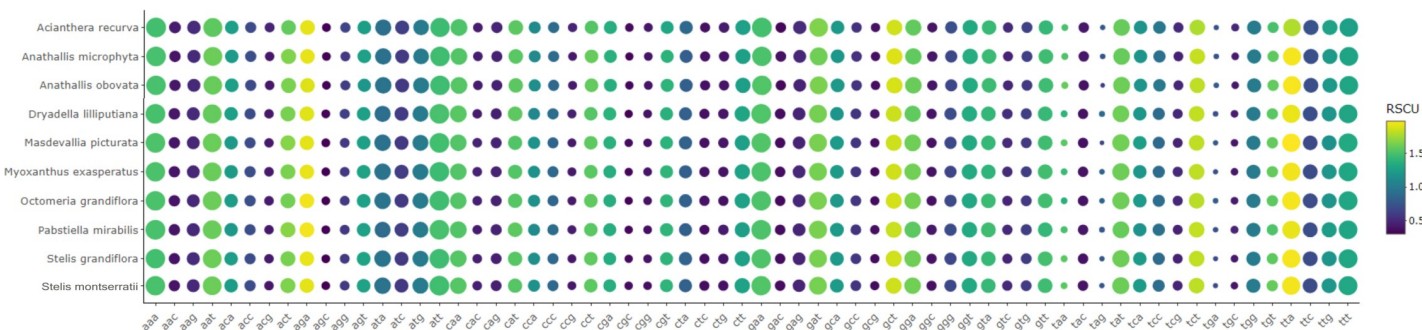

**Fig 2. Heatmap of relative synonymous codon usage (RSCU) and codon relative frequency observed for Pleurothallidinae CDSs.** The lowest frequency is indicated by purple and highest yellow. Relative frequencies are proportional to circle sizes.

of which 170 (13%) are unique and varied 113–143 per species (S8 Table). They were most common in the LSC (71%) and IGSs (55.6%) and least abundant in the IR (13.5%) and introns (16.2%). This distribution is as expected: the LSC is the largest region [22], and IGSs are the most common and variable plastome category [22, 72]. This is similar to what was found in *Bulbophyllum* [29] and *Dendrobium* [27] plastomes. Mono- and trinucleotide repeats are the most common and hexanucleotides the rarest, the latter absent in some species (S8 Table). As for other Orchidaceae [27, 29, 68, 73], A/T repeats predominate, followed by AAG/CTT. Most trinucleotide microsatellites were found in CDSs and the IR, the latter possessing nearly all of these. Indeed, tri- and hexanucleotide SSRs were previously reported to be more frequent in CDSs than any other plastome category [74], undoubtedly due to maintenance of protein function [74, 75].

Plastid microsatellites are excellent tools for genotyping, genetic mapping, and population studies [76–78]. Therefore, we designed 54 primers that cover 62 polymorphic SSR loci present in at least seven Pleurothallidinae (S9 Table). These repeats consist of 17 polyTs (10–19x), nine polyAs (10–19x), three di- (4–23x), 31 tri- (3–10x), one tetra- (3x), and one pentanucleotide repeat (3–4x). We hope that providing this set of microsatellite primers will encourage more population and species delimitation studies in Pleurothallidinae.

## Mutation hotspots

We analyzed levels of variation in 104 sequences more than 150 bp long, comprising IGSs, introns, and two CDSs (S10 Table). The most variable sequences (higher SV) are IGSs that have low GC content (Fig 3). This inverse relation between SV and %GC was expected and previously reported in *Bulbophyllum* and *Dendrobium*, as AT-rich sequences have higher mutation rates [27, 29, 68]. Seven of the ten sequences with highest SV found (*ndhF-rpl32*, *psbB-psbT*, *psbK-psbI*, *rpl16-rps3*, *rpl32-trnL*, *trnR-atpA*, and *trnS-trnG*) were previously identified in various other orchid clades [24, 26–29, 64, 68], but we identify here for the first time in orchids the following: *petN-psbM*, *psbI-trnS*, and *trnW-trnP*.

Most phylogenetic studies in Pleurothallidinae have used nrITS, sometimes combined with the *trnK*$^{UUU}$ intron or *matK* (*e.g.* [6, 14]). Other markers such as *ycf1* and *trnH-psbA* were included more recently [20, 21]. These all have SV below 25%, in contrast with 29.63–41.67% for the ten hotspots (S10 Table). Therefore, we designed subtribe-specific primers for the top ten to assist future phylogenetic analyses in Pleurothallidinae (S11 Table).

## Phylogenomics

The aligned matrix of Epidendreae plastome sequences consisted of 142,507 bp, ~10% of which are variable and of these 41.47% are potentially parsimony-informative (PIC). As expected, non-coding regions had more variable characters (13.69%) and proportionally fewer PICs, whereas protein-coding loci are more conservative (Table 3) [79]. Tree topology based on plastid hotspots was the same as plastid non-coding DNA (NC-DNA) analysis (Figs 4 and S5). When comparing ML trees from the plastome datasets, the only disagreement was the position of *Masdevallia picturata*, but not with strong support (Fig 4).

Based on our results (Fig 4), *Octomeria grandiflora* (*Octomeria* affinity sensu Karremans [18]) is sister to the rest of the sampled subtribe, followed next by the clade of *Acianthera recurva* (*Acianthera* affinity) and *Myoxanthus exasperatus* (*Restrepia* affinity). Then *Dryadella lilliputiana* (*Specklinia* affinity) is sister to the last two clades. The first is composed of *Anathallis microphyta* and *A. obovata* (*Lepanthes* affinity), and the second by *Pabstiella mirabilis* sister to *Stelis grandiflora* and *S. montserratii* (*Pleurothallis* affinity). *Masdevallia picturata* (*Masdevallia* affinity) is sister to the *Pleurothallis* affinity clade in whole plastome and CDS analyses,

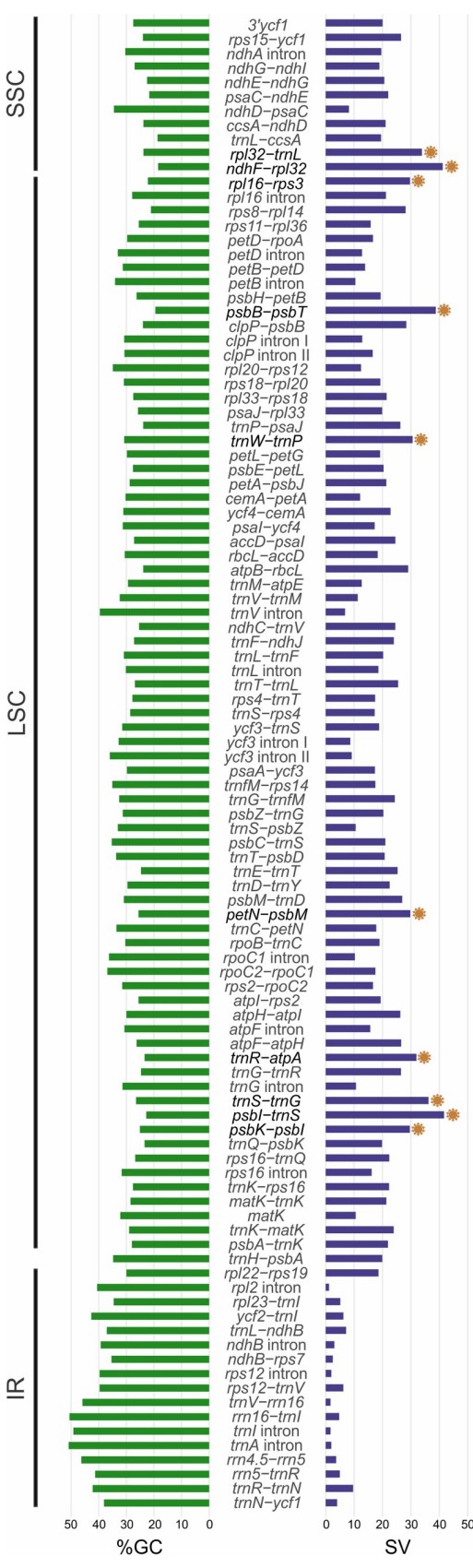

**Fig 3. Graph of sequence variability (SV) and %GC for 104 Pleurothallidinae plastid sequences.** The ten most-variable sequences are highlighted.

and sister to the *Lepanthes* affinity clade in NC-DNA and hotspots analyses, with higher support in the latter.

Comparing this topology with Karremans' proposal [18], the positions of the *Octomeria*, *Restrepia*, and *Acianthera* affinities in this order agree. However, the *Masdevallia* and *Specklinia* affinities were not positioned accordingly. The classification of Pleurothallidinae genera into affinities is a compilation of phylogenetic results based almost exclusively on nrITS data, so this disagreement may be due to genomic incongruence. In agreement, the nuclear ribosomal DNA (nrDNA) operon analysis recovered the *Specklinia+Pleurothallis* relationship (S6 Fig). The mtDNA analysis, on the other hand, recovered only the *Lepanthes* and *Acianthera+Myoxanthus* clades, with the former sister of the rest (S6 Fig). It must be noted that such a sparse sample of genera does not constitute a robust evaluation of Karremans' ideas of relationships, and our aim was to describe and compare the large amount of molecular data presented here.

Plastid molecular markers are widely used in phylogenetic studies of land plants due to their abundance in cells and easy amplification/sequencing [19, 80, 81]. The nuclear ribosomal internal transcribed spacers (nrITS) have poorer sequencing success than plastid markers, but they hold the majority of nrDNA operon molecular variability and have great discriminant power at the species level for most angiosperms [19, 82]. Hence, plastid markers are often combined with nrITS, specially in orchid phylogenetic studies (e.g. [6, 14, 21, 25, 83, 84]). Highly supported discordance between plastid and nuclear trees is uncommon in Orchidaceae but has been detected in Epidendroideae [55], especially in Catasetinae [85] and here in

**Table 3. Number of sequences, length, nucleotide variation, and best substitution model for each dataset and partitions.**

| Dataset | # sequences | Length (bp) | Variable characters | PIC | Best model (AIC) |
|---|---|---|---|---|---|
| **Whole plastomes** | 12 | 142,507 | 14,003 (09.82%) | 5,807 (41.47%) | GTR+F+R2 |
| | 9 | 139,059 | 11,045 (07.94%) | 3,330 (30.15%) | GTR+F+R2 |
| **Coding sequences** | 12 | 58,979 | 3,672 (06.22%) | 1,596 (43.46%) | GTR+F+R2 |
| **Non-coding sequences** | 12 | 76,868 | 10,526 (13.69%) | 3,971 (37.72%) | TIM+F+R2 |
| **Hotspots** | 12 | 6,656 | 1,676 (25.18%) | 717 (42.78%) | Partitioned |
| *ndhF-rpl32* | 12 | 1,146 | 370 (32.28%) | 135 (36.48%) | TVM+F+G4 |
| *petN-psbM* | 12 | 1,042 | 231 (22.17%) | 102 (44.15%) | TIM+F+G4 |
| *psbB-psbT* | 12 | 611 | 170 (27.82%) | 66 (38.82%) | K3Pu+F+G4 |
| *psbI-trnS$^{GCU}$* | 12 | 165 | 45 (27.27%) | 21 (46.67%) | GTR+F |
| *psbK-psbI* | 12 | 617 | 127 (20.58%) | 54 (42.52%) | TVM+F+G4 |
| *rpl16-rps3* | 12 | 199 | 48 (24.12%) | 25 (52.08%) | TVM+F+I |
| *rpl32-trnL$^{UAG}$* | 12 | 988 | 260 (26.31%) | 116 (44.61%) | TVM+F+R2 |
| *trnR$^{UCU}$-atpA* | 12 | 231 | 59 (25.54%) | 25 (42.37%) | K3Pu+F+G4 |
| *trnS$^{GCU}$-trnG$^{UCC}$* | 12 | 1,456 | 319 (21.91%) | 147 (46.08%) | K3Pu+F+G4 |
| *trnW$^{CCA}$-trnP$^{UGG}$* | 12 | 201 | 47 (23.38%) | 26 (55.32%) | K3Pu+F+R3 |
| **mtDNA** | 9 | 94,117 | 6,048 (06.43%) | 899 (14.86%) | TVM+F+I |
| **nrDNA operon** | 9 | 5,865 | 335 (05.71%) | 152 (45.37%) | GTR+F+R2 |
| **Combined**$^*$ | 9 | 239,041 | 17,506 (07.32%) | 4,381 (25.68%) | Partitioned |

PIC = parsimony-informative characters.

$^*$Plastome + mtDNA + nrDNA operon.

## Plastomes

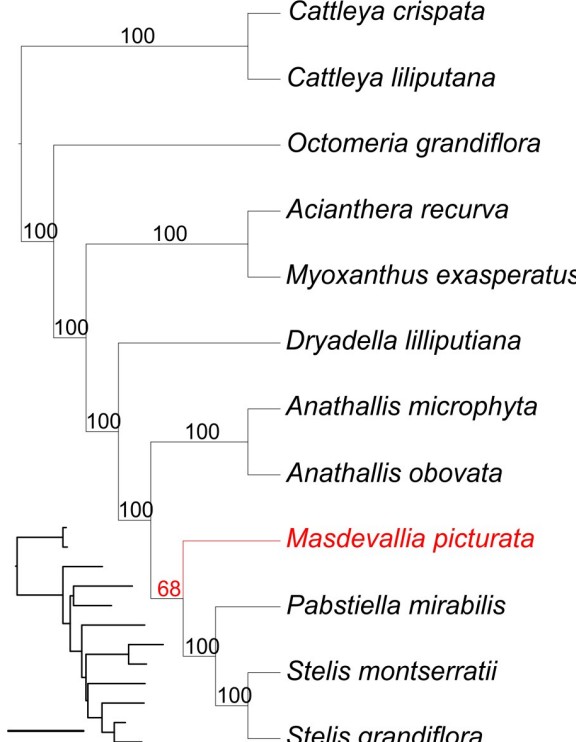

## Plastid hotspots

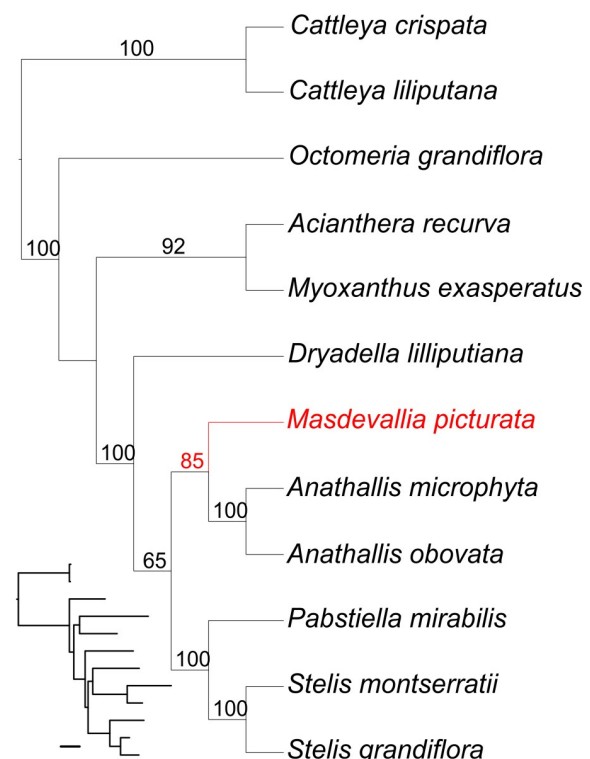

## Combined
### (Plastomes + mtDNA + nrDNA operon)

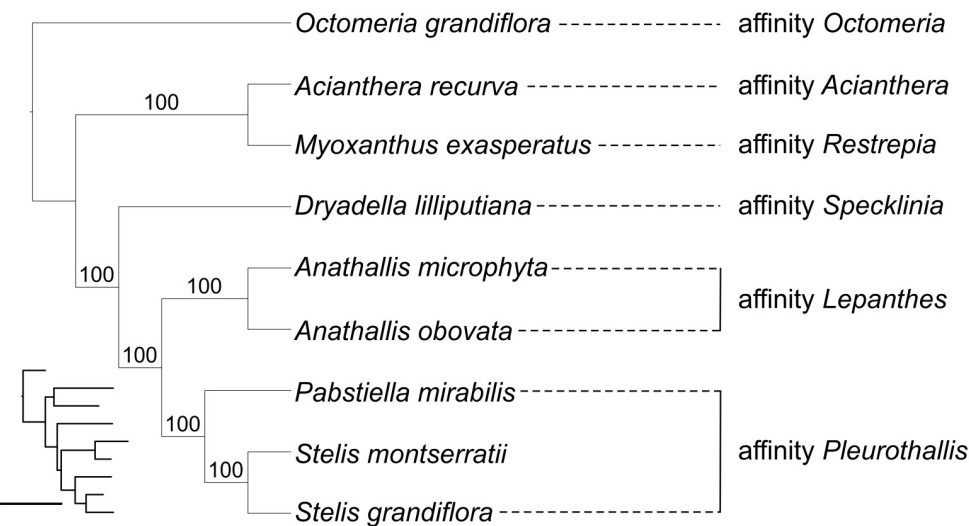

**Fig 4. Maximum likelihood trees based on complete plastomes (one IR), plastid hotspots, and combined analyses (plastomes (one IR) + mtDNA + nrDNA operon).** Numbers on branches are the bootstrap percentages; a tree with proportional branch lengths is on the left of each tree, which bars represent 0.02 nucleotide substitutions per site.

Pleurothallidinae. Such cytonuclear discordance can be due to divergence in evolutionary history between nuclear and plastid genomes and is usually attributed to introgression, incomplete lineage sorting, and hybridization [85–88]. Determining the causes of the observed cytoplasmic discordance is beyond our scope due to limited sampling, but these newly sequenced plastomes are a good starting point for future investigations on this subject in the Pleurothallidinae.

Mitochondrial molecular markers were not considered as potential plant DNA barcodes due to their low substitution rates [19, 89, 90]. Complete mitochondrial genome sequences are also difficult to assemble in plants because their overall structure is not as conserved as those of plastomes and contains horizontally inherited nuclear and plastid sequences [91]. Still, some mitochondrial markers have been analyzed in the past for use in orchid analyses [92], such as *nad1* and *cox1* introns [93, 94], but their low variability compared to plastid markers discouraged their use. More recently, several mitochondrial CDSs were included in a phylogenomic study in the family [95], which produced a highly supported tree with some incongruence with the plastid analysis. Our mtDNA analysis included mostly CDSs and partial introns and also produced a well-supported tree, but it significantly diverged from the plastid and nuclear results (S6 Fig). It is noteworthy that there is no orchid complete mitochondrial genome available, so we recovered the mtDNA from raw reads based on a reference sequence that is phylogenetically distant from Pleurothallidinae. In addition, sequencing for each species varied in read number, length, and quality, which directly impacted mtDNA length and content (Table 2). We believe that these two main factors generated high sequence heterogeneity among species that might have influenced the results, so they must be interpreted with care. Despite the odd topology, the mtDNA tree retrieved the *Lepanthes* affinity and *Acianthera* +*Myoxanthus* clades found in other analyses. Also, a high-throughput analysis for the *Lepanthes horrida* Rchb.f. species complex revealed that some mitochondrial markers were among the ten best performing markers [96], supporting a future role in Pleurothallidinae phylogenetics.

When the three datasets were combined (plastome, nrDNA operon, and mtDNA), we got the same topology as the plastome analysis with higher bootstrap percentages (Fig 4). In fact, applying standard methods to concatenated multigene data often improves tree resolution and support in phylogenetic estimates (e.g. as seen in Pleurothallidinae [21, 96]). Simulation-based studies have shown that phylogenetic accuracy increases as more genes are included to the dataset [97, 98]. With this in view, we hope that the molecular markers highlighted here as plastid hotspots will be used in future phylogenetic studies in Pleurothallidinae in conjunction with nrITS, mtDNA, and, if possible, with morphological and ecological data as well, so that more robust phylogenetic trees are produced.

## 4. Conclusions

The eight new plastomes of Pleurothallidinae sequenced here greatly increased orchid subtribal representation in GenBank. Their overall structure and codon usage are conserved, and gene content is similar, with variation only in the *ndh* gene composition. Protein-coding genes under positive selection were detected, and a complete set of primers were provided for microsatellites and the top ten most-variable markers, thus increasing molecular resources available to future evolutionary research in the subtribe at various scales, from genes to species. In particular, the top ten markers are more variable than any plastid markers previously used in Pleurothallidinae phylogenetics, and their tree topology is similar to that obtained with whole plastomes, reinforcing their potential as suitable molecular markers for the subtribe. However, strongly supported incongruence among plastid, nuclear ribosomal DNA, and mtDNA

topologies suggests putative divergence in the evolutionary histories of these genomes, a topic that needs future investigation. The well-supported mitochondrial tree, together with the performance in recent studies, suggests that the inclusion of mitochondrial markers in phylogenetic studies of Pleurothallidinae could be useful. The combined analysis of the three genomes, in addition to improving the support, could be expected to circumvent partly the problems associated with individual analyses.

## Supporting information

**S1 Fig. Genetic maps of the Pleurothallidinae plastomes sequenced.** Genes are represented by rectangles, for which functions are identified by colors as shown in the legend. Genes placed inside the circle are transcribed clockwise, and those outside the circle are transcribed counter-clockwise. The gray inner circle is the GC content graph. Images of the sequenced species were taken by Eric C. Smidt except Marcelo Rodrigues for *Myoxanthus exasperatus*.
(TIF)

**S2 Fig. Graph of *ndh* gene content of the ten Pleurothallidinae plastomes analyzed.** White squares = complete gene losses, light-grey squares = truncated reading frames (pseudogenes), and dark-grey squares = complete CDSs.
(TIF)

**S3 Fig. Mauve alignment of the ten Pleurothallidinae plastomes analyzed.**
(TIF)

**S4 Fig. Correlation plots.**
(TIF)

**S5 Fig. Maximum likelihood trees based on Epidendrae plastid CDS and non-coding DNA (NC-DNA).** Numbers on branches are the bootstrap percentages; a tree with proportional branch lengths is on the left of each tree, which bars represent 0.02 nucleotide substitutions per site.
(TIF)

**S6 Fig. Maximum likelihood trees based on Pleurothallidinae plastomes (one IR), nuclear ribosomal DNA operon, and mitochondrial DNA.** Numbers on branches refer to bootstrap percentages; a tree with proportional branch lengths is on the left of each tree, which bars represent 0.02 nucleotide substitutions per site.
(TIF)

**S1 Table. Taxonomic information, voucher, and GenBank accession numbers of all sequences used.** Accessions in boldface are sequences generated in this study. All vouchers provided are deposited at UPCB herbarium.
(PDF)

**S2 Table. NGS information of the eight plastomes sequenced and the mtDNA and nuclear ribosomal DNA recovered.** *See Mauad *et al.* [30].
(PDF)

**S3 Table. Gene content of the mitochondrial DNA recovered from raw reads.** *Genes with introns.
(PDF)

**S4 Table. Gene content of the Pleurothallidinae genomes analyzed.** [a]Genes with introns, [b]duplicated genes (in IRs), [c]partially duplicated genes, *pseudogenes.
(PDF)

**S5 Table. Number of insertion/deletion events by genomic region in each Pleurothallidinae plastome.** The plastome of *Madevallia picturata* was the reference for all indels.
(PDF)

**S6 Table. Relative frequency (RF) and relative synonymous codon usage (RSCU) of all 64 codons and their respective amino-acids (AA) in each Pleurothallidinae plastome analyzed.** Species names were abbreviated as follows: ACIRE = *Acianthera recurva*, ANAMI = *Anathallis microphyta*, ANAOB = *A. obovata*, DRYLI = *Dryadella lilliputiana*, MASPI = *Masdevallia infracta*, MYOEX = *Myoxanthus exasperatus*, OCTGR = *Octomeria grandiflora*, PABMI = *Pabstiella mirabilis*, STEGR = *Stelis grandiflora*, and STEMO = *Stelis montserratii*.
(PDF)

**S7 Table. Results of Tajima's neutrality test (D) for the 68 CDSs common to the 10 Pleurothallidinae plastomes analyzed.** D > 0 = purifying selection, and D < 0 = positive selection. Significant results (p ≤ 0.05) were highlighted in bold. *Tajima's D test was not computed due to the lack of variable sites.
(PDF)

**S8 Table. Characterization, distribution, and frequency of microsatellites (SSRs) in the Pleurothallidinae plastomes analyzed.**
(PDF)

**S9 Table. List of the 62 polymorphic microsatellites present in at least seven Pleurothallidinae plastomes, including anticipated size, primers, melting temperatures, and location.** SSR type in brackets with repetition frequency following. Loci amplified by the same primer separated by semicolons. Slanting bars (/) indicate polymorphism in repeat units.
Tm = primer melting temperature.
(PDF)

**S10 Table. Nucleotide sequences of Pleurothallidinae with more than 150 bp in the alignment sorted by sequence variability (SV), including their location in the plastome, length, and guanine-cytosine content (GC).** In total, 104 sequences were analyzed, comprising intergenic spacers (IGS), introns, and popular molecular markers for Orchidaceae such as the *trnH-psbA* IGS, the *matK* CDS, and the 3' portion of *ycf1* CDSs. IGS = inter-genic spacer, CDS = protein-coding sequence.
(PDF)

**S11 Table. Primers for the ten most variable regions for Pleurothallidinae.** SV = sequence variability, Tm = primer melting temperature. *Primers previously published but with mismatches in Pleurothallidinae. For these cases, subtribe-specific primers for the same regions are provided.
(PDF)

**S1 File. Codes and datasets used in R analyses.**
(ZIP)

**S2 File. Misa-web results for all ten Pleurothallidinae plastomes analyzed.**
(ZIP)

**S3 File. Alignment matrices of Epidendrae (complete plastomes with one IR) and Pleurothallidinae (complete plastomes with one IR, mtDNA, and nrDNA).**
(ZIP)

## Acknowledgments

We thank the Universidade Federal do Paraná for providing the infrastructure for the realization of this study. AVSRM especially thanks Michelle Zavala Páez for helping with the statistical analyses and the reviewers for improving the manuscript.

## Author Contributions

**Conceptualization:** Anna Victoria Silvério R. Mauad.

**Data curation:** Anna Victoria Silvério R. Mauad, Leila do Nascimento Vieira, Valter Antônio de Baura, Eduardo Balsanelli.

**Formal analysis:** Anna Victoria Silvério R. Mauad.

**Funding acquisition:** Eric de Camargo Smidt.

**Investigation:** Anna Victoria Silvério R. Mauad.

**Methodology:** Anna Victoria Silvério R. Mauad, Valter Antônio de Baura, Eduardo Balsanelli, Emanuel Maltempi de Souza.

**Project administration:** Eric de Camargo Smidt.

**Resources:** Emanuel Maltempi de Souza, Eric de Camargo Smidt.

**Software:** Anna Victoria Silvério R. Mauad.

**Supervision:** Leila do Nascimento Vieira, Eric de Camargo Smidt.

**Validation:** Mark W. Chase, Eric de Camargo Smidt.

**Visualization:** Mark W. Chase, Eric de Camargo Smidt.

**Writing – original draft:** Anna Victoria Silvério R. Mauad, Eric de Camargo Smidt.

**Writing – review & editing:** Anna Victoria Silvério R. Mauad, Leila do Nascimento Vieira, Mark W. Chase, Eric de Camargo Smidt.

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
