## [Decision Letter · Decision Letter 0]

8 Apr 2021

PONE-D-21-05263

Phylogenomic study in Pleurothallidinae (Orchidaceae): conservative plastomes, new variable markers, and comparative analyses among plastid, nuclear, and mitochondrial data

PLOS ONE

Dear Dr. Silvério Righetto Mauad,

Thank you for submitting your manuscript to PLOS ONE. After careful consideration, we feel that it has merit but does not fully meet PLOS ONE’s publication criteria as it currently stands. Therefore, we invite you to submit a revised version of the manuscript that addresses the points raised during the review process.

We look forward to receiving your revised manuscript.

Kind regards,

Zhong-Jian Liu

Academic Editor

PLOS ONE

Journal Requirements:

2. In your Methods, please provide details of the source of all plant material used in your study, and details of any permits obtained if any material was collected from the wild.

3. We note you have included a table to which you do not refer in the text of your manuscript. Please ensure that you refer to Table 7 in your text; if accepted, production will need this reference to link the reader to the Table.

Additional Editor Comments (if provided):

Mauad et al presented the phylogenomic study of Pleurothallidinae using molecular markers and comparative analyses among molecular data. However, both reviewers raise concerns over some methods and writings in the manuscript. Please carefully revise the manuscript based on both reviewers' comments and improve the language too.

Reviewers' comments:

Reviewer's Responses to Questions

**Comments to the Author**

1. Is the manuscript technically sound, and do the data support the conclusions?

Reviewer #1: Partly

Reviewer #2: Yes

2. Has the statistical analysis been performed appropriately and rigorously? 

Reviewer #1: No

Reviewer #2: Yes

3. Have the authors made all data underlying the findings in their manuscript fully available?

Reviewer #1: Yes

Reviewer #2: Yes

4. Is the manuscript presented in an intelligible fashion and written in standard English?

Reviewer #1: No

Reviewer #2: Yes

5. Review Comments to the Author

Reviewer #1: This study sequenced plastomes of eight species in subtribe Pleurothallidinae, and also carried out comparative analysis of among plastid, nuclear, and mitochondrial data. There is too much figures and tables in the body part of the manuscript, and some figure and table can shift to supplementary section, such as Table 4, Table 5, Table 6, Table 7, Table 8. And Table 1 and Table 2 can combine in one table.

Furthermore, Tajima’s D test is mainly used in population genetics, the result of positive selection in this study might contain false positives.

Also, there are minor errors in the paper:

L121 Is “CLC Genomics” indicate “CLC Genomics Workbench” ?

L212-L213 The classification of the bootstrap is unreasonable.

L295 “Zhitao et al.” should be “Niu et al.”

L577-L579 the citation of Reference 27 is incorrect

Fig. 1, the plastome structure of the eight species are quite similar, the plastome map of one species is enough.

Fig. 6, the tree topology of the four datasets is identical, one is enough.

Reviewer #2: The authors presented a phylogenomic study in Pleurothallidinae (Orchidaceae) with eight newly plastid genomes. The plastid phylogenomic tree was also compared with nuclear ribosomal DNA operon and partial mitochondrial genome sequences. I believe this study comprises important information that is worth to be published. However, the study requires considerable work to be publishable.

1. I do not recommend the combined analysis based on plastome, nuclear ribosomal DNA operon and mitochondrial DNA (Fig. 7). Plastome and nuclear ribosomal DNA operon is subject to gene conversion, making it effectively uniparentally inherited (although there is no guarantee that the same parental copy will be kept in nuclear ribosomal DNA operon), whereas mitochondrial DNA could exhibit two alleles, making interpretation of results sometimes challenging. In addition, the incongruences have been shown between plastome and nuclear ribosomal DNA operon, e.g., Dryadella. I recommend the combined analysis should be based on the incongruence length difference (ILD) test.

2. I would like to suggest put the Figs. 2,3, 5,6, into Supporting Information, the same as Tables 1, 3, 5, 6, 7, 8, 9. It is difficult to read in body, so many figures and tables. The writing should be more focus on the topic or key, rather than everything moved out.

3. Further the writing can be improved. Many sentences are unclear and would benefit from being rephrased. I marked several sentences that need to be rephrased in the abstract, but stopped doing so for the rest of the manuscript. e.g., Line 26, please check the “first”, the plasomes of Anathallis obovata and Masdevallia picturata have been reported in previous study. The research background should be stated.

6. PLOS authors have the option to publish the peer review history of their article (what does this mean?). If published, this will include your full peer review and any attached files.

Reviewer #1: No

Reviewer #2: No

---

## [Author Response · Author response to Decision Letter 0]

18 May 2021

Editor:

R.: The manuscript was thorougly revised to meet PLOS ONE's style requirements, as well as the Figures and file naming.

2. In your Methods, please provide details of the source of all plant material used in your study, and details of any permits obtained if any material was collected from the wild.

R.: Details provided.

3. We note you have included a table to which you do not refer in the text of your manuscript. Please ensure that you refer to Table 7 in your text; if accepted, production will need this reference to link the reader to the Table.

R.: All tables and figures are now correctly cited in the text.

Reviewer #1:

1. There is too much figures and tables in the body part of the manuscript, and some figure and table can shift to supplementary section, such as Table 4, Table 5, Table 6, Table 7, Table 8. And Table 1 and Table 2 can combine in one table.

R.: The revised manuscript now contains only four figures and three tables. Former Figs 1 and 2 and Tables 1, 3, 4, 5, 6, and 7 moved to Supplementary Material. Former Figs 6 and 7 split into Fig 4 and S5 and S6 Figs.

2. Furthermore, Tajima’s D test is mainly used in population genetics, the result of positive selection in this study might contain false positives.

R.: Tajima’s D test computes the difference between the average number of nucleotide differences and the number of segregating sites, which is expected to not exist in neutrality (the evolution rate is the same as the mutation rate), or, in other words, in the absence of selection. Indeed, the test is widely used in population genetics, but since it only needs DNA polymorphism to assess neutrality we do not think it is inappropriate for use on our data. Besides, most of the CDSs under selection according to our results were also reported in several other phylogenomic studies to be under selection in Orchidaceae. It is true that there might be false positives, but this is true in all cases in which it is used. Our conclusion in this case is simply that these regions test positive for selection, and because we do not use this result in any additional calculations or conclusions we maintain that our use is appropriate and correctly interpreted.

3. Also, there are minor errors in the paper:

L121 Is “CLC Genomics” indicate “CLC Genomics Workbench” ?

R.: Yes, and all mentioned in the text were corrected to CLC Genomics Workbench.

4. L212-L213 The classification of the bootstrap is unreasonable.

R.: Bootstrap percentages are no longer classified into “weak, moderate or strong”, but the 95% limit used instead as proposed by Feselstein. Citation also substituted.

5. L295 “Zhitao et al.” should be “Niu et al.”. L577-L579 the citation of Reference 27 is incorrect.

R.: Citation corrected.

6. Fig. 1, the plastome structure of the eight species are quite similar, the plastome map of one species is enough.

R.: Fig. 1 shifted to Supplementary Material.

7. Fig. 6, the tree topology of the four datasets is identical, one is enough.

R.: Actually, the Epidendreae trees diverged in the position of Masdevallia picturata, and all Pleurothallidinae trees diverged in some way, so we thought it was important to show all topologies. Fig 4 (former Fig 6) now presents the Epidendreae plastome and hotspot trees to show (1) that the variation in hotspots represents well the whole plastomes and (2) the divergence in M. picturata positioning and also the Pleurothallidinae combined tree. All other trees moved to Supplementary Material.

Reviewer #2:

1. I do not recommend the combined analysis based on plastome, nuclear ribosomal DNA operon and mitochondrial DNA (Fig. 7). Plastome and nuclear ribosomal DNA operon is subject to gene conversion, making it effectively uniparentally inherited (although there is no guarantee that the same parental copy will be kept in nuclear ribosomal DNA operon), whereas mitochondrial DNA could exhibit two alleles, making interpretation of results sometimes challenging. In addition, the incongruences have been shown between plastome and nuclear ribosomal DNA operon, e.g., Dryadella. I recommend the combined analysis should be based on the incongruence length difference (ILD) test.

R.: Plastomes and mitochondrial genomes are mostly maternally inherited (in most cases, as in Pleurothallidinae), and thus they do not have more than one allele per individual (except for heteroplasmy, which is not the case here). Plastome sequences are not subject to gene conversions, so this reviewer is incorrect in this assertion. These genes are present in single non-recombining molecule (one chromosome). Our molecular data are perfectly combinable, and the incongruence found is presented and discussed, although the cause of disagreement is beyond the scope of this paper. If we tried to hide the incongruence or did not mention it, then there would be a problem. Plus, many other studies combine nuclear and plastid DNA to obtain more robust phylogenetic hypotheses despite gene tree incongruence. The ILD is not necessary when incongruence is clear and strongly supported. 

2. I would like to suggest put the Figs. 2,3, 5,6, into Supporting Information, the same as Tables 1, 3, 5, 6, 7, 8, 9. It is difficult to read in body, so many figures and tables. The writing should be more focus on the topic or key, rather than everything moved out.

R.: This issue was also indicated by Reviewer #1. However, we do not agree with the suggestion that the writing was not focused or that everything was moved out to the figures and tables. The figures and tables provided only illustrated what was discussed in the text, and that is the reason why we agree in moving most of the figures and tables to the Supplementary Material.

3. Further the writing can be improved. Many sentences are unclear and would benefit from being rephrased. I marked several sentences that need to be rephrased in the abstract, but stopped doing so for the rest of the manuscript. e.g., Line 26, please check the “first”, the plasomes of Anathallis obovata and Masdevallia picturata have been reported in previous study. The research background should be stated.

R.: We did not receive any tracked-changed text, so we were not sure of what sentences should be rephrased and why. Nevertheless, we revised the entire manuscript carefully, including by the native English speaker author. We also provided the research background for the plastome sequences as required.

---

## [Decision Letter · Decision Letter 1]

24 Jun 2021

PONE-D-21-05263R1

Plastid phylogenomics of Pleurothallidinae (Orchidaceae): conservative plastomes, new variable markers, and comparative analyses of plastid, nuclear, and mitochondrial data

PLOS ONE

Dear Dr. Silvério Righetto Mauad,

Thank you for submitting your manuscript to PLOS ONE. After careful consideration, we feel that it has merit but does not fully meet PLOS ONE’s publication criteria as it currently stands. Therefore, we invite you to submit a revised version of the manuscript that addresses the points raised during the review process.

We look forward to receiving your revised manuscript.

Kind regards,

Zhong-Jian Liu

Academic Editor

PLOS ONE

Journal Requirements:

Additional Editor Comments (if provided):

This version has addressed most of the questions raised by both reviewers. While there are some minor mistakes, please carefully check and revise the typos throughout the manuscript.

Reviewers' comments:

Reviewer's Responses to Questions

**Comments to the Author**

1. If the authors have adequately addressed your comments raised in a previous round of review and you feel that this manuscript is now acceptable for publication, you may indicate that here to bypass the “Comments to the Author” section, enter your conflict of interest statement in the “Confidential to Editor” section, and submit your "Accept" recommendation.

Reviewer #1: All comments have been addressed

Reviewer #2: All comments have been addressed

2. Is the manuscript technically sound, and do the data support the conclusions?

Reviewer #1: Partly

Reviewer #2: Yes

3. Has the statistical analysis been performed appropriately and rigorously? 

Reviewer #1: Yes

Reviewer #2: Yes

4. Have the authors made all data underlying the findings in their manuscript fully available?

Reviewer #1: Yes

Reviewer #2: Yes

5. Is the manuscript presented in an intelligible fashion and written in standard English?

Reviewer #1: Yes

Reviewer #2: Yes

6. Review Comments to the Author

Reviewer #1: There are many minor errors in the revised version, e.g.

Line 107：“grouos” should be changed to “groups”

Line 448: “invvestigation” should be changed to “investigation"

Minor errors in the Reference part, such as 20, 29, 79, 80, 82, 83

Reviewer #2: This manuscript is much improved. The authors have addressed my concerns as well as can be expected.

7. PLOS authors have the option to publish the peer review history of their article (what does this mean?). If published, this will include your full peer review and any attached files.

Reviewer #1: No

Reviewer #2: No

---

## [Author Response · Author response to Decision Letter 1]

28 Jun 2021

Reviewer #1: There are many minor errors in the revised version, e.g.

Line 107：“grouos” should be changed to “groups”

Line 448: “invvestigation” should be changed to “investigation"

Minor errors in the Reference part, such as 20, 29, 79, 80, 82, 83.

R.: Line 107: “grouos” corrected to “groups”. Line 448: “invvestigation” corrected to “investigation”. Minor erros on references 20, 29, 46, 79, 80, 82, and 83, such as word spacing, spelling, author names, journal abbreviation, and punctuation marks were corrected.

---

## [Editor Report · Decision Letter 2]

30 Jul 2021

Plastid phylogenomics of Pleurothallidinae (Orchidaceae): conservative plastomes, new variable markers, and comparative analyses of plastid, nuclear, and mitochondrial data

PONE-D-21-05263R2

Dear Dr. Silvério Righetto Mauad,

We’re pleased to inform you that your manuscript has been judged scientifically suitable for publication and will be formally accepted for publication once it meets all outstanding technical requirements.

Kind regards,

Zhong-Jian Liu

Academic Editor

PLOS ONE

Additional Editor Comments (optional):

This revision basically addressed all the questions raised by both reviewers. Therefore, acceptance is recommended.
---

## [Editor Report · Acceptance letter]

20 Aug 2021

PONE-D-21-05263R2 

Plastid phylogenomics of Pleurothallidinae (Orchidaceae): conservative plastomes, new variable markers, and comparative analyses of plastid, nuclear, and mitochondrial data 

Dear Dr. Silvério R. Mauad:

I'm pleased to inform you that your manuscript has been deemed suitable for publication in PLOS ONE. Congratulations! Your manuscript is now with our production department. 

Kind regards, 

on behalf of

Professor Zhong-Jian Liu 

Academic Editor

PLOS ONE